# Transarterial Bleomycin–Lipiodol Chemoembolization for the Treatment of Giant Hepatic Hemangiomas: An Assessment of Effectiveness

**DOI:** 10.3390/cancers16020380

**Published:** 2024-01-16

**Authors:** Arkadiusz Kacała, Mateusz Dorochowicz, Adrian Korbecki, Michał Sobański, Michał Puła, Dariusz Patrzałek, Dariusz Janczak, Maciej Guziński

**Affiliations:** 1Department of General, Interventional and Neuroradiology, Wroclaw Medical University, 50-367 Wrocław, Poland; 2Faculty of Medicine, Wroclaw Medical University, 50-367 Wrocław, Poland; m.dorochowicz97@gmail.com; 3Department of General, Interventional and Neuroradiology, Wroclaw University Hospital, 50-556 Wrocław, Poland; lek.adriankorbecki@gmail.com (A.K.); michal.pula97@gmail.com (M.P.); 4Department of Vascular, General and Transplantation Surgery, Wroclaw Medical University, 50-367 Wrocław, Poland; dariusz.patrzalek@umw.edu.pl (D.P.); dariusz.janczak@umw.edu.pl (D.J.)

**Keywords:** giant hepatic hemangioma, transcatheter arterial chemoembolization (TACE), bleomycin–lipiodol emulsion, liver tumor treatment, interventional radiology

## Abstract

**Simple Summary:**

The article explores an innovative treatment for large liver tumors known as giant hepatic hemangiomas. These tumors, although typically benign, can cause significant health issues and symptoms, necessitating effective treatment methods. This study focuses on the use of a specific technique called superselective transcatheter arterial chemoembolization (TACE), utilizing a bleomycin–lipiodol emulsion. The researchers aimed to assess the efficacy of this method in reducing the size of these tumors and alleviating the associated symptoms. The findings of this research are important because they provide insights into a less-invasive alternative to surgical treatment for giant hepatic hemangiomas. This study demonstrates that this technique has a high rate of technical success, with a significant reduction in tumor size observed in the majority of cases. Importantly, this approach also appears to have a manageable safety profile. The results of this study could have a significant impact on the medical community’s approach to treating giant hepatic hemangiomas, offering a potentially safer and more effective treatment option than traditional surgical methods.

**Abstract:**

This study evaluates the effectiveness of superselective transcatheter arterial chemoembolization (TACE) using a bleomycin–lipiodol emulsion in treating giant hepatic hemangiomas. A retrospective review included 31 patients with a mean age of 53 ± 10.42 years who underwent TACE from December 2014 to October 2022, with follow-up imaging examinations to assess outcomes. Technical success was defined as successful embolization of all feeding arteries, and clinical success was defined as a reduction in hemangioma volume by 50% or more on follow-up imaging. This study observed a 100% technical success rate. Post-embolization syndrome was common, and two cases of asymptomatic hepatic artery dissection were noted. Clinical success was achieved in 80.6% of patients, with significant volume reduction observed in the majority. Conclusively, superselective transcatheter arterial chemoembolization with bleomycin–lipiodol emulsions is presented as a viable and effective treatment option for giant hepatic hemangiomas. With no procedure-related mortality and significant volume reduction in most cases, this method offers a promising alternative to surgical intervention. This study’s findings suggest a need for further exploration and validation in larger-scale prospective studies.

## 1. Introduction

Hepatic hemangiomas are the most common benign liver tumor, with the reported prevalence ranging from 0.4% to 20% in the general population. This type of lesion predominantly occurs in women in their fourth and fifth decade of life [1,2] (shown in Figure 1).

In most instances, hepatic hemangiomas are small and asymptomatic, and patients maintain normal liver function, leading to the incidental discovery of the tumor during imaging for unrelated conditions [3,4,5,6,7,8]. Typically, conservative management and monitoring suffice for such cases [6,9,10]. Lesions exceeding 5 cm in diameter are termed giant hemangiomas and are more likely to cause symptoms, necessitating treatment [3,11]. Although the management of hepatic hemangiomas is subject to debate, there is a growing consensus that treatment is warranted for symptomatic, rapidly enlarging tumors or for those posing a significant risk of rupture and hemorrhage [12,13].

The most common symptoms include abdominal pain, nausea, and early satiety, potentially due to thrombosis, infarction, intralesional bleeding, capsular distension, or the pressure exerted on adjacent organs by the enlarged liver [14,15]. Particularly large hemangiomas may lead to serious complications such as hemorrhage, local compression, Kasabach–Merritt syndrome, or Budd–Chiari syndrome [14,16,17,18]. The standard surgical removal of symptomatic hemangiomas typically involves open or laparoscopic techniques [5,19,20,21,22,23,24], with common postoperative complications including blood loss, bile leakage, ileus, and wound infection. The associated morbidity and mortality rates are reported to be 13–21% and 0–2%, respectively [3,25,26,27,28,29]. Consequently, TACE has been increasingly adopted as an effective alternative therapy for hepatic hemangiomas, although previous research has indicated variability in its efficacy and adverse event profile [30,31]. This study aimed to retrospectively evaluate the effectiveness of transarterial bleomycin–lipiodol chemoembolization in patients with giant hepatic hemangiomas.

## 2. Materials and Methods

In this single-center, retrospective study, 73 patients with giant hepatic hemangiomas underwent TACE utilizing a bleomycin–lipiodol emulsion from December 2014 to October 2022. The focus was on the subset of 31 patients for whom follow-up imaging data were available. The inclusion criteria encompassed (1) patients who received chemoembolization for giant hepatic hemangiomas at our facility from 2014 to 2022, (2) patients with a confirmed diagnosis of hepatic hemangiomas via imaging techniques such as computed tomography (CT) or magnetic resonance imaging (MRI), and (3) patients with hemangiomas measuring over 5 cm in diameter. Patients with an atypical angiographic appearance of the hemangioma on arteriography were excluded.

Image assessments were performed to ascertain the size, number, and liver involvement of the hemangiomas. Informed consent was duly obtained from all participants. Comprehensive blood counts, liver function tests, and coagulation profiles were conducted prior to TACE to confirm patient eligibility for the procedure. Clinical success was determined as a reduction in hemangioma volume of 50% or more on follow-up imaging. Technical success was defined as the successful embolization of all arteries feeding the hemangioma.

Post-chemoembolization, angiography was carried out to assess the precise drug coverage at the hemangioma borders. This assessment was critical for determining the efficacy of the procedure and the distribution of the drug within the lesion. A four-grade scale, as proposed by a previous study, was utilized to evaluate lesion coverage, offering a systematic and standardized method for assessing the dispersion of the bleomycin–lipiodol mixture [31].

The grading of drug coverage was pivotal for determining the distribution of the bleomycin–lipiodol mixture around the hemangioma’s periphery. Grade 1 indicated that the therapeutic agents covered less than 25% of the rim. Grade 2 signified coverage extending from 25% to 50% of the rim. Grade 3 denoted that the agents covered up to 75% of the rim but did not provide complete coverage. Finally, Grade 4 signified complete coverage of the rim by the therapeutic agents [31]. Utilizing this detailed grading system allowed for an accurate assessment of the extent of drug dispersion within the hemangioma. This, in conjunction with the results from follow-up imaging, offered valuable insights into the effectiveness and thoroughness of the chemoembolization treatment.

Radiation doses were meticulously measured and recorded for each procedure to ensure patient safety. The treatment’s safety was rigorously assessed during the patients’ hospital stay. This included close monitoring of patients for the occurrence and severity of post-embolization syndrome, which is characterized by symptoms such as abdominal pain, fever, nausea, and fatigue. To quantify the intensity of pain in patients experiencing post-embolization syndrome, a detailed four-grade verbal rating scale was employed. The scale was defined as 0 = no pain, 1 = mild pain, 2 = moderate pain, and 3 = severe pain. Additionally, all patients were thoroughly monitored during the postoperative period to identify any serious adverse events that might have arisen during or after the intervention. The mortality rate of patients both during the procedure and throughout the hospital stay was also considered a critical safety metric. By enacting these exhaustive evaluation protocols, we aimed to prioritize and protect the safety and well-being of patients at all stages of the treatment process.

Following local anesthesia, the common femoral artery was punctured with an 18 G needle, and a 5F femoral sheath was inserted. A 5F Simon 1 or Cobra 2 catheter facilitated precise navigation through the celiac trunk and the superior mesenteric artery. Arteriography was performed to accurately identify the primary feeding artery of the hemangioma. Then, a 2.4 Fr or 2.7 Fr Progreat (Terumo, Tokyo, Japan) microcatheter was advanced into the feeding artery of the hemangioma. Under fluoroscopic guidance, a solution containing 1500 IU of bleomycin (Pfizer Inc., New York, NY, USA) and 7–15 cc of lipiodol (Guerbet, Villepinte, France) was mixed using standard three-way stopcocks and slowly injected into the hemangioma until the desired distribution was achieved. The maximum bleomycin dose per session was limited to 15,000 IU; the amount of lipiodol was tailored to the hemangioma’s size, not to exceed 30 mL per session. The distribution of bleomycin–lipiodol within the hemangioma was evaluated using control hepatic arteriography post-embolization utilizing a grading scale that assessed coverage of the hemangioma’s borders. To gauge hemangioma size reduction, CT or MRI was performed, on average 12.5 ± 11.3 months after the procedure (range 1.4–47.67 months). Significant tumor shrinkage was defined as a reduction of 50% or more in the volume of the hemangioma, as measured using follow-up imaging. Patients were subsequently considered for further TACE sessions, if necessary, based on tumor size reduction or symptom recurrence.

Statistical analysis was run with R software (version 4.1.2). Nominal variables were characterized with no. of counts and % share in the group. Numeric variables are described with the mean (SD) or median (IQR), depending on distribution normality. Distribution normality was verified with the Shapiro–Wilk test as well as skewness and kurtosis. Variance homogeneity was assessed with the Levene test. Groups were compared using Student’s t, t-Welch, or Mann–Whitney U tests for numeric variables and chi-square or Fisher’s exact test for nominal variables, as appropriate. Relationships between two numeric variables were verified using Pearson or Spearman correlation analysis, depending on distribution normality. Two-step logistic regression was used to identify factors for clinical success. Variable selection to multivariate model was first based on *p* < 0.25 (univariate regression) criterion [32]. Furthermore, a stepwise approach was employed to select variables for the final model. Multivariate model fit was verified with Nagelkerke R2 and the Hosmer and Lemeshow GOF test. Collinearity was checked with VIF indicators. All statistical tests assumed alpha = 0.05.

## 3. Results

A total of 31 patients—25 women and 6 men—were included in this study. The mean age of the participants was 53 ± 10.42 years, ranging from 33 to 72 years. Prior to the procedure, all patients had experienced symptoms such as nausea, abdominal distension, abdominal pain, dyspepsia, or early satiety at varying levels of severity. Of the lesions assessed, 13 were located in the left lobe of the liver, with volumes ranging from 51.91 cm^3^ to 2828.8 cm^3^; 17 were in the right lobe, with volumes ranging from 64.77 cm^3^ to 1269.45 cm^3^; and 1 spanned both lobes, with a volume of 973.44 cm^3^. This distribution provides valuable insights into the anatomical involvement and precise locations of the hemangiomas, thereby improving our understanding of the disease and the potential challenges associated with its treatment. The most common preoperative volume of the hemangiomas (54.8%) was between 100 cm^3^ and 500 cm^3^; 16.1% of the lesions were smaller, and 29% were larger.

Regarding the number of procedures required, 45.2% of patients underwent a single procedure, 32.3% needed a second procedure, 16.1% required three interventions, suggesting more complex cases, and 6.5% underwent four procedures, indicating a resistant condition necessitating multiple treatments. All procedures were performed at the same center, ensuring consistency and a standardized approach to patient management. The baseline characteristics of the patients, including age, sex, and relevant medical history, were collected and are summarized in Table 1, displaying the mean values along with their standard deviations.

TACE was performed with a technical success rate of 100%. Post-embolization syndrome was observed in 50.9% of the procedures on the first day, with abdominal pain being the most common symptom, occurring in 96.5% of cases at varying levels of severity. Regarding hemangioma coverage, grade 4 (>75%) was achieved in 75.4% of cases, followed by grades 3 and 1 (each at 8.8%), and grade 2 (7%). The mean follow-up period was 12.5 ± 11.3 months, ranging from 1.4 to 47.67 months.

In this study, a significant majority of patients, 25 out of 31 (80.6%), achieved clinical success, defined as a regression in hemangioma volume of more than 50% compared to preoperative imaging (shown in Figure 2).

The reduction in hemangioma volumes pre- and post-treatment was statistically significant, with a *p*-value of <0.001, affirming the procedure’s efficacy. The most common range of volume reduction was 70–80%, which occurred in 32.26% of cases, and a substantial regression of greater than 90% was the second most frequent outcome, observed in 25.81% of patients. These findings highlight the high effectiveness of the procedure, as not only was clinical success (volume regression >50%) achieved in the majority of cases, but a more significant reduction exceeding 70% was observed in 67.74% of all cases. The mean volume regression recorded in this study was 72.26%, further demonstrating the significant impact of the treatment. The detailed range of hemangioma volume regression, indicative of the procedure’s effectiveness, is presented in Table 2.

## 4. Discussion

The etiology of hepatic hemangiomas is not fully understood. Structurally, hemangiomas are composed of venous lacunes lined by vascular endothelial cells and are separated by connective tissue walls. These lesions slow the internal blood flow primarily supplied by the hepatic artery [33]. The overwhelming majority of cavernous hemangiomas are asymptomatic and small lesions that do not compromise liver function and are often incidentally discovered during ultrasound imaging [34,35,36]. Typically, this type of hemangioma does not necessitate any intervention aside from conservative management and surveillance. Nevertheless, factors such as steroid or estrogen therapy and pregnancy might precipitate an increase in hemangioma size, yet there have been no documented cases of malignant transformation in hepatic hemangiomas [37,38].

Giant liver hemangiomas (greater than 5 cm) can lead to serious and potentially life-threatening symptoms and complications, including persistent abdominal pain, vascular compression, intra-hepatic or intra-abdominal bleeding, gastric outlet obstruction, obstructive jaundice, Kasabach–Merritt syndrome, or Budd–Chiari syndrome [16,29,39,40,41]. The emergence of symptoms, rapid growth, or a high risk of hemangioma rupture calls for alternative therapeutic strategies beyond the conventional ‘resection versus observation’ approach [10,29,42,43]. Historically, surgery has been the standard treatment for symptomatic patients or those with significant growth [44,45].

However, operative management of hepatic hemangiomas is associated with significant risks of surgical complications, particularly in patients with comorbidities, and therefore must be approached with caution. Morbidity rates ranging from 10 to 27% and mortality rates as high as 2% following enucleation or liver resection have been reported [27,28].

Before considering interventional procedures, it is crucial to rule out other potential causes of symptoms such as reflux disease, peptic ulcer disease, or gallstones. Schnelldorfer et al. indicated that the short-term morbidity risks associated with surgical treatment are akin to the long-term adverse effect risks of nonoperative treatments [42].

Formerly, various treatment techniques appeared to be valid alternatives to the surgical approach, including radiofrequency ablation (RFA) and microwave ablation (MWA). However, these were concluded to be ineffective, particularly for tumors with a diameter smaller than 7 cm [46,47,48]. Moreover, the occurrence of severe complications from RFA, such as systemic inflammatory response syndrome (SIRS) and hemolysis-related complications, has been reported to be as high as 34–100% in tumors larger than 10 cm in diameter [49,50]. Hepatic hemangiomas, consisting of endothelial cells derived from the hepatic artery, suggest that treatments targeting vessel blockage may be effective [3,7,51]. Transarterial embolization (TAE) has emerged as a viable approach for managing hepatic hemangiomas. Nevertheless, there exists a lack of consensus concerning the efficacy of TAE in treating hemangiomas and the potential severity of the associated complications [30,52,53,54,55,56,57]. To further increase hemangioma volume reduction by inhibiting blood vessel growth, chemotherapeutic agents can be combined with TAE, known as TACE. These treatments can be used either preoperatively to minimize intraoperative blood loss [8] or as a definitive treatment [3,51,58], and their non-invasive nature results in favorable cosmetic outcomes. However, Liu et al. found the utilization of TAE with pingyangmycin in combination with lipiodol for liver hemangiomas to be unsatisfactory, posing a considerable risk of severe complications [54]. Among 55 patients, 4 individuals (7.3%) experienced severe complications, such as biloma and bile duct necrosis leading to abscess formation. In a 5-year follow-up, only 19 out of 53 patients (35.8%) observed a reduction in hemangioma size or maintained the same size as before undergoing TAE. Conversely, Torkian et al. reported an opposing view in a systematic review and meta-analysis, stating that TAE with various agents, such as bleomycin, pingyangmycin, or ethanol, in combination with lipiodol, was safe and effective [30]. The study reported no instances of mortality, and CIRSE grade 3 complications were observed in only 6 out of 1450 cases (0.4%). The clinical response to TAE was documented to range between 63.3% and 100%.

Bleomycin is a cycle-nonspecific cytotoxic agent with antimicrobial, angiosclerotic, and antiangiogenic properties, commonly utilized in the treatment of vascular anomalies, such as percutaneous hemangiomas [59,60]. It gradually elicits a generalized inflammatory response around the tumor and within the portal area [61]. Lipiodol, an iodized oil, can be combined with bleomycin, enhancing its embolic effect and serving as a carrier to improve distribution directly to the targeted area. It accumulates in the blood sinuses and can remain within lesions for up to a year, thereby controlling lesion size reduction [62]. Transarterial bleomycin–lipiodol embolization effectively eradicates the pathological vascular bed through the gradual destruction of endothelial cells [61].

Yuan et al. conducted a retrospective review of the clinical and radiological outcomes in patients with giant hepatic hemangiomas, concluding that TAE using a bleomycin–lipiodol emulsion is both safe and effective [57]. Their study reported no mortality or symptomatic recurrence, with a reduction rate of over 50% at 6 months in 88.1% of cases. In the current study, the technical success of TACE was 100%, with no mortality observed. Furthermore, Yuan et al.’s observation of a significant correlation between volume regression and the time from procedure to follow-up was corroborated [57].

For giant hepatic hemangiomas, the benefits of treatment and subsequent shrinkage are considerable. As the lesion grows, it increasingly encroaches on liver volume, sometimes significantly encumbering the volume of healthy liver parenchyma. The reduction in the lesion alleviates pressure on the liver, allowing the previously occupied space to be reclaimed by the liver parenchyma, potentially restoring its normal function.

Severe complications resulting from transarterial bleomycin–lipiodol embolization, such as hepatic artery dissection, cholecystitis, liver failure, intrahepatic hemorrhage, hepatic infarction, intrahepatic biloma, and splenic infarction, have been reported [29,63]. However, the prevalence of these severe complications has mostly been documented in isolated case reports or as a minority within broader studies, suggesting their rare occurrence [53]. In the present study, two instances of asymptomatic hepatic artery dissection were detected during the angiography conducted at the end of the procedure. Although these cases were asymptomatic, their identification highlights the importance of careful monitoring and thorough post-procedural evaluations to swiftly identify and manage any potential complications.

Bile duct complications are infrequent, occurring in only 0.87–4% of cases [64,65,66]. The biliary ducts have a distinct vascular supply. While the normal hepatic parenchyma receives a dual blood supply, with 75–80% coming from the portal vein and 20–25% from the hepatic artery, the primary blood supply to the bile ducts primarily originates from the branches of the hepatic artery, forming a dual capillary network that encircles the bile ducts [67,68]. During transarterial chemoembolization (TACE), there is a risk that the hepatic arteries or arterioles supplying the bile duct may undergo embolization, leading to ischemic necrosis of the bile duct. The optimal approach for managing bile duct injuries following TACE has not been conclusively established. Previous reports have suggested that the mild dilation of intrahepatic bile ducts may not require specific treatment, and regular clinical follow-up is sufficient [69,70]. Bilomas may naturally resolve without intervention. However, cases of growing bilomas accompanied by jaundice may require biliary drainage to alleviate biliary obstruction. In our study, we did not notice any biliary complications, as superselective embolization allows for the targeted embolization of only those vessels that go directly to the tumor.

Pulmonary fibrosis has been reported as a potential complication in patients treated with particularly high cumulative doses of bleomycin (300 mg) [71]. Throughout this study, the maximum dose of bleomycin administered per session was 15 mg, substantially below this threshold. The most frequent adverse event was TACE-related post-embolization syndrome, characterized by symptoms akin to influenza, such as fever, abdominal pain, nausea, and vomiting; these symptoms rarely necessitated intervention beyond analgesics [15,72,73,74,75]. Post-embolization syndrome was observed following 50.9% of the procedures in this study, with pain being the most prevalent symptom, reported in 96.5% of cases.

The study’s retrospective nature and the lack of biopsy preceding the diagnosis due to the typically benign nature of the lesion and its characteristic features on CT/MRI imaging are limitations. Additionally, of the 73 patients with giant hemangiomas treated with TACE at our center, long-term clinical and imaging follow-ups were only available for 31 patients. Many asymptomatic individuals declined follow-up visits, further constraining the scope of our findings. Consequently, additional prospective studies with extended follow-ups are required to more accurately evaluate the efficacy of transarterial bleomycin–lipiodol embolization in patients with giant hepatic hemangiomas.

## 5. Conclusions

This study demonstrates that superselective transcatheter arterial chemoembolization using bleomycin–lipiodol emulsions is a viable and effective treatment option for giant hepatic hemangiomas. With a technical success rate of 100% and no procedure-related mortality, the treatment led to significant volume reduction in the majority of the cases. Clinical success, defined as a volume reduction of more than 50%, was achieved in 80.6% of patients, with the most common range of volume reduction being 70–80%. These promising outcomes highlight the potential of TACE as a less invasive alternative to surgery, offering a high rate of efficacy with manageable safety concerns. As such, TACE with bleomycin–lipiodol emulsions represents a significant advancement in the therapeutic approach to managing giant hepatic hemangiomas, warranting further exploration and validation in larger-scale prospective studies.

## Figures and Tables

**Figure 1 cancers-16-00380-f001:**
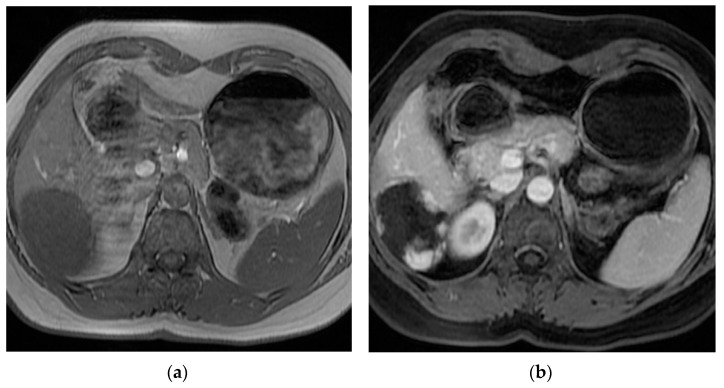
Abdomen MRI scan of a patient with giant hepatic hemangioma.: (**a**) pre-contrast T1; (**b**) post-contrast T1 FS.

**Figure 2 cancers-16-00380-f002:**
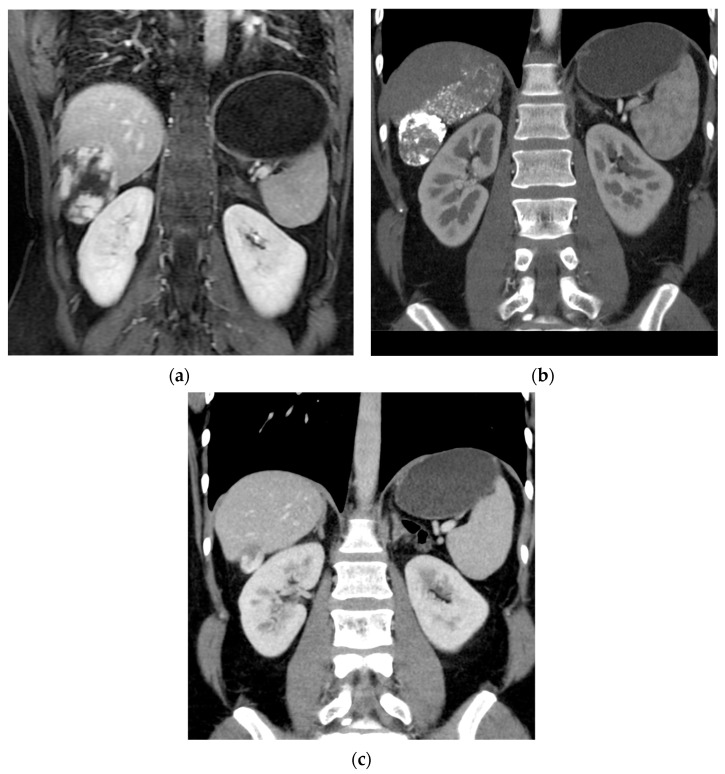
Reduction in size of a giant hepatic hemangioma in the right lobe following embolization: (**a**) pre-embolization MRI; (**b**) follow-up CT at 1 year post-embolization; (**c**) follow-up CT at 2 years post-procedure.

**Table 1 cancers-16-00380-t001:** Patients’ basic characteristics (*n* = 31).

Variables	Group	Frequency	Percent
Sex	Male	6	80.6
	Female	25	19.4
Age	<40 years	2	6.5
	40–49 years	11	35.5
	50–60 years	10	32.3
	>60 years	8	25.8
Location	Left lobe	13	41.9
	Right lobe	17	54.8
	Both lobes	1	3.2
Drug coverage grade	<25%	5	8.8
	25–50%	4	7
	50–75%	5	8.8
	>75%	43	75.4
Number of procedures	1	14	45.2
	2	10	32.3
	3	5	16.1
	4	2	6.5
Preoperative volume	<100 cm^3^	5	16.1
	100–500 cm^3^	17	54.8
	500–1000 cm^3^	7	22.6
	>1000 cm^3^	2	6.5
Post-embolization syndrome	Yes	29	50.9
	No	28	49.1

**Table 2 cancers-16-00380-t002:** Volume shrinkage (*n* = 31).

Group	Frequency	Percent
30–40%	2	6.45
40–50%	4	12.9
50–60%	1	3.12
60–70%	3	9.68
70–80%	10	32.26
80–90%	3	9.68
>90%	8	25.81

## Data Availability

Data available on request.

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
