# Peer review of "Transarterial Bleomycin–Lipiodol Chemoembolization for the Treatment of Giant Hepatic Hemangiomas: An Assessment of Effectiveness"

_cancers, 2024, doi:10.3390/cancers16020380_

Round 1

Reviewer 1 Report

Comments and Suggestions for Authors

Manuscript ID:cancers-2801271

Manuscript Title:Transarterial Bleomycin-Lipiodol Chemoembolization for the Treatment of Giant Hepatic Hemangiomas: An Assessment of Effectiveness

Manuscript Type:Article

Comments: Transarterial Bleomycin-Lipiodol Chemoembolization for the Treatment of Giant Hepatic Hemangiomas: An Assessment of Effectiveness are widely recognized, but there is also debate, especially regarding complications of the bile duct. The author reported their results. The subject of this manuscript is of value, but there are a few of defects need to be modified.

1. Should the author briefly discuss the the prevalence, causes, and prevention of bile duct complications (cholecystitis and intrahepatic biloma) based on literature.

2. The terms or phrases used throughout the text should be expressed consistently. Please check the following terms.Bleomycin-Lipiodol,Bleomycin-Lipiodol,lipiodol-bleomycin......

3. I found that the author has published the following related manuscript, did the contents of these two manuscripts overlap: KacaÅ‚a A, Dorochowicz M, PatrzaÅ‚ek D, Janczak D, GuziÅ„ski M. Safety and Feasibility of Transarterial Bleomycin-Lipiodol Embolization in Patients with Giant Hepatic Hemangiomas. Medicina (Kaunas). 2023 Jul 25;59(8):1358. doi: 10.3390/medicina59081358. PMID: 37629648; PMCID: PMC10456525.

4. Abbreviations are typically defined the first time the term is used within the abstract and again in the main text and then used exclusively throughout the remainder of the document. Please consider adhering to this convention. Please check: Transarterial Chemoembolization (TACE) ,computed tomography (CT) ,magnetic resonance imaging (MRI)...... 

Author Response

Dear Reviewer,

Thank you for your valuable feedback on our manuscript titled "Transarterial Bleomycin-Lipiodol Chemoembolization for the Treatment of Giant Hepatic Hemangiomas: An Assessment of Effectiveness." We have addressed your comments as detailed below:

  1. Bile Duct Complications: In response to your suggestion, we have added a comprehensive paragraph in the discussion section of our article. This new addition discusses the prevalence, causes, and prevention of bile duct complications such as cholecystitis and intrahepatic biloma, incorporating relevant literature. This inclusion aims to provide a more thorough understanding of these complications in the context of Transarterial Bleomycin-Lipiodol Chemoembolization.

  2. Consistency in Terms: We have thoroughly reviewed and revised the manuscript to ensure consistent use of terms. All variations of "Bleomycin-Lipiodol" have been standardized to maintain uniformity and clarity throughout the text.

  3. Overlap with Previous Manuscripts: Our previously published work primarily addressed the safety and feasibility of Transarterial Bleomycin-Lipiodol Embolization. While there is some thematic overlap, the current manuscript specifically focuses on the effectiveness of this treatment for giant hepatic hemangiomas. We have ensured that the content of this manuscript provides additional insights and is distinct from our previous publication.

  4. Abbreviations: We have revised the manuscript to clearly define all abbreviations, such as TACE, CT, and MRI, at their first occurrence in both the abstract and the main text. These abbreviations are then consistently used throughout the document.

We believe these revisions comprehensively address your concerns and enhance the manuscript's overall quality. We appreciate the opportunity to refine our work based on your insightful feedback.

Reviewer 2 Report

Comments and Suggestions for Authors

This is an interesting study of a radiologic technique for the treatment of giant hemangiomata.

It is well presented and the conclusions are supported by the results. It is disappointing that a significant number of patients who had the procedure did not have follow up imaging but one would presume that they felt relatively well post procedure and needed no further intervention.

I do not think that Figure 3 is particularly useful.

Author Response

Dear Reviewer,

Thank you for your thoughtful and constructive review of our manuscript on the radiologic technique for the treatment of giant hemangiomas. We appreciate your positive remarks about the presentation and the support of our conclusions by the results.

Regarding your feedback on Figure 3, we have removed this figure from the manuscript as per your suggestion. We understand and agree that it might not have added significant value to the study's findings.

Additionally, we acknowledge your point regarding the lack of follow-up imaging for a notable number of patients. As you rightly mentioned, we also presume that these patients experienced satisfactory outcomes post-procedure, reducing the perceived need for further intervention. In future studies, we aim to implement a more rigorous follow-up protocol to ensure comprehensive data collection and analysis.

We are grateful for your insights, which have helped us improve the quality and coherence of our manuscript.